# Piercing and Surface-Crack Defects in Cold Combined Forward-Backward Extrusion

**Heng-Sheng Lin [1],\*, Chien-Yu Lee [2] and Wen-Shun Li [1]**

[1] Department of Mold and Die Engineering, National Kaohsiung University of Science and Technology, Kaohsiung 807, Taiwan; eric6906727@yahoo.com.tw

[2] General Education Center, Taiwan Shoufu University, Tainan 721, Taiwan; leechanyo@gmail.com

\* Correspondence: hslin@nkust.edu.tw

**Featured Application: A forming limit diagram of the combined forward-backward extrusion (CFBE) process is developed to provide a conception in choosing appropriate extrusion ratios in forming fasteners featuring a forward extruded pin and a backward extruded cup.**

**Abstract:** Metal flow tends to be complex and difficult to predict in the combined forward-backward extrusion (CFBE) process. Piercing and surface-crack defects are phenomenal in forming fasteners featuring a forward extruded pin and a backward extruded cup. In this work, a series of the CFBE tests with various combinations of the forward extrusion ratio (FER) and the backward extrusion ratio (BER) were conducted. A forming limit diagram, detailed with the piercing and surface-crack defects on the forward extruded pin or the backward extruded cup, was developed to provide a conception in choosing appropriate extrusion ratios in forming fasteners with such pin-and-cup features. With the aid of the forming load-stroke curves and the finite element analysis of fracture damage, the fracturing mechanism for the CFBE process was provided.

**Keywords:** combined forward-backward extrusion; piercing defect; surface-crack defect; fracturing damage

## 1. Introduction

The cold extrusion operation, which is an essential scheme of the cold forging process, has been broadly used in forming fasteners [1] and automotive parts [2], while the hot extrusion process is widely used in forming lightweight sections/profiles [3–5]. Figure 1 shows the cross-section of a support pin [1] with a crack defect by cold extrusion. The inner side of the extruded cup has to be threaded for its final usage. Figure 2 shows the sequence of a cold forging part [2] with cracks appearing on the side surface during the second forming scheme. The extruded rod was thread-rolled in the separate stage.

There are forward and backward extrusions [6] in the application of cold extrusion. The so-called combined forward-backward extrusion (CFBE) process, when the two unidirectional extrusion schemes are used simultaneously, is shown in Figure 3. The metal flow tends to be complex and difficult to predict in the CFBE. Piercing and surface-crack defects are phenomenal and to be investigated in this work.

Yoon et al. [7] investigated the characteristics of warm forming of AZ31B magnesium alloy with the CFBE process. They found that the crack was initiated and developed easily under the low static pressure condition. They also concluded that the low static pressure was attributed to the fact that the size of the forward extruded part was close to that of the backward extruded cup. Yoon et al. [8] further examined the effects of process parameters on the forming characteristics of AZ31B magnesium alloy in CFBE at warm temperatures. They concluded that the extrusion load increases as the forming temperature decreases, or as the punch speed increases, or backward extrusion ratio (BER) becomes large.

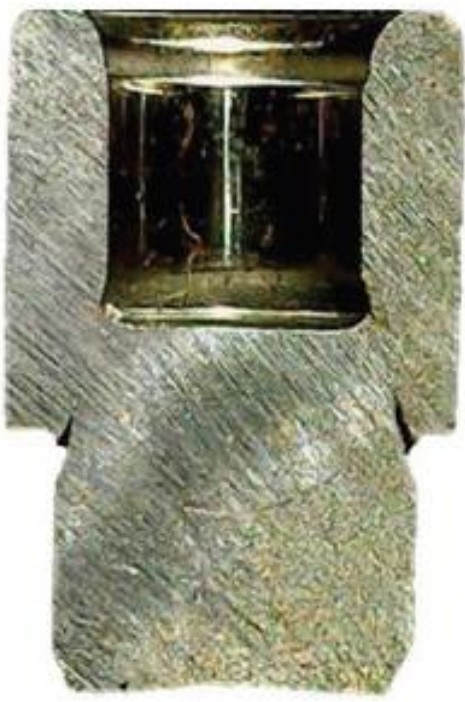

**Figure 1.** A crack defect of support pin by cold combined forward-backward extrusion (CFBE) [1].

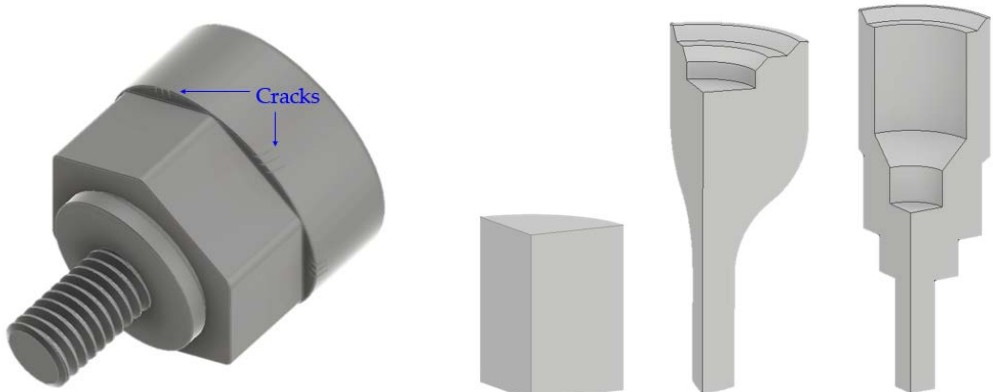

**Figure 2.** (**a**) Cracks appearing on side surface of a cold forged part; (**b**) sequence of cold forging operations [2].

In our previous work on the forming of a support pin [1] with the CFBE process, surface-cracks occurred on the extruded part and sometimes piercing defects were observed. The size of the forward extruded pin was not close to that of the backward extruded cup, but the surface-cracks and even piercing defects still occurred. This outcome indicates that the fracturing mechanism of the CFBE has not been fully comprehended. Therefore, in this work a series of the CFBE tests with various combinations of the forward extrusion ratio (FER) and the backward extrusion ratio (BER) were conducted. A forming limit diagram, detailed with the defects on the forward extruded pin or the backward extruded cup, was developed to provide a conception in choosing appropriate extrusion ratios in forming fasteners with such pin-and-cup features. In particular, the finding of the forming defects on the backward extruded cup has not been reported from the literature.

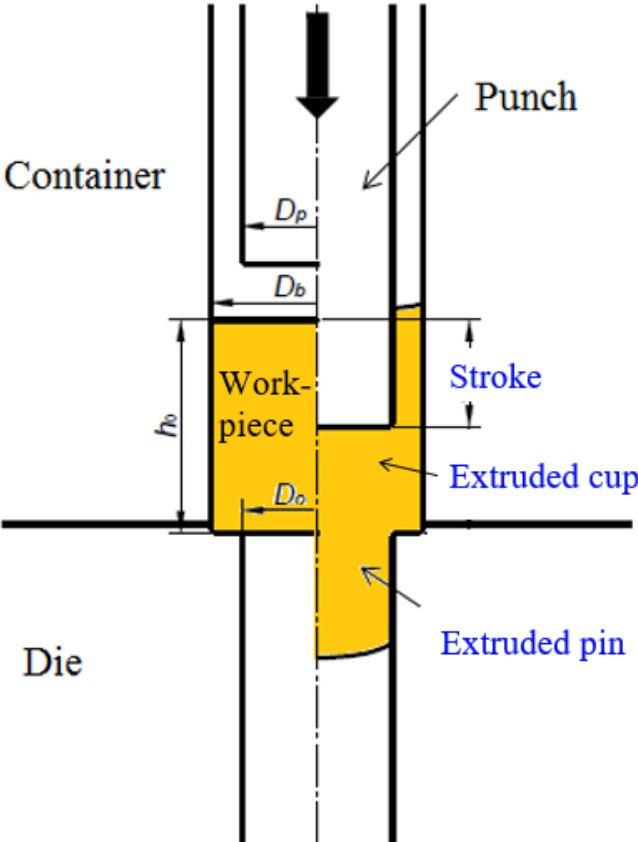

**Figure 3.** Configuration of CFBE process, left: before extrusion; right: after extrusion.

## 2. Materials and Methods

In this study, annealed ASTM A36 structural steel billets with a diameter of 25 mm and a height of 25 mm were used to investigate the forming defects of the CFBE process. The Young's modulus was 200 GPa, and the Poisson ratio was 0.26. The CFBE experiments were conducted on a hydraulic press at 25 °C. The punching speed was 10 mm/s, and solid $MoS_2$ powder was used as the lubricant.

Because the current CFBE as shown in Figure 3 is categorized as a bulk metal forming process and axis-symmetric in nature, a commercial finite element software DEFORM 2D was used for the analysis. The reason why DEFORM was chosen was because it uses solid elements and is suitable in investigating the fracture problems caused by the large plastic deformation from CFBE. Other named software such as AutoForm and DynaForm use shell elements and were suitable for the sheet-metal forming process, while ANSYS and Abaqus are appropriate in the simulation of elastic deformation or small plastic deformation.

The workpiece was assumed to be plastic and the punch and die were assumed to be rigid, and this was attributed to the current investigation focusing on the cause of the fracture due to the large deformation from metal forming. The elastic behavior such as springback from the workpiece and the punch and die was, therefore, neglected to facilitate the simulation efficiency. The flow rule of the material was:

$$\sigma = 850\varepsilon^{0.25} \tag{1}$$

which was obtained by curve fitting the tensile test of the work billets. There were 15,000 elements used in the simulation and the friction factor was assumed to be 0.12. The effects of strain rate and temperature on the flow rule were neglected.

There is no ductile damage criterion that is satisfactory for most metal-forming processes [2]. The normalized Cockcroft and Latham damage criterion [9] is the most widely used due to its simplicity. The DEFORM software provides a built-in sub-routine and is to

be used in determining the level of ductile fracture, and helps "scope" the occurrence of the forming defects. The normalized Cockcroft and Latham damage criterion is defined as:

$$C = \int_0^{\bar{\varepsilon}_f} \frac{\sigma^*}{\bar{\sigma}} d\bar{\varepsilon}, \tag{2}$$

where C is the critical damage value, $\sigma^*$ is the maximum tensile stress, $\bar{\sigma}$ is the effective stress, $\bar{\varepsilon}$ is the effective strain, and $\bar{\varepsilon}_f$ is the critical effective strain upon material fracture, respectively. The corresponding critical damage value upon fracture was 1.3, which was obtained from matching the tensile test of the work billets.

Table 1 lists the mechanical properties of the workpiece and Table 2 lists the parameters used in the simulation. Both the forward extrusion ratio and the backward extrusion ratio are expressed as:

$$\text{FER} = \frac{\pi D_b^2 / 4}{\pi D_o^2 / 4} \tag{3}$$

and,

$$\text{BER} = \frac{\pi D_b^2 / 4}{\pi D_b^2 / 4 - \pi D_p^2 / 4} \tag{4}$$

respectively. $D_b$, $D_p$ and $D_o$ are the diameters of the billet, punch and die outlet, respectively. The FER ranged from 1.33 to 4.00 and the BER ranged from 1.20 to 4.00. Punches of various diameters ($D_p$) and die inserts of various outlet diameters ($D_o$) had to be replaced in order to accommodate the combination of the FER and BER values used in the CFBE tests.

**Table 1.** Mechanical properties of the workpiece.

| Mechanical Properties | Values |
|---|---|
| Yield stress (MPa) | 351.4 |
| Tensile strength (MPa) | 613.5 |
| Young's modulus (GPa) | 200 |
| Poisson ratio | 0.26 |
| Elongation (%) | 41.2 |

**Table 2.** Parameters used in simulation.

| Parameters | Values |
|---|---|
| Workpiece | Plastic |
| Punch and die | Rigid |
| Flow rule | $\sigma = 850\varepsilon^{0.25}$ |
| Elements | 15,000 |
| Friction factor | 0.12 |
| Temperature (°C) | 25 |
| Punching speed (mm/s) | 10 |
| Lubricant | $MoS_2$ |
| Damage criterion | Normalized Cockcroft and Latham |
| FER (forward extrusion ratio) | 1.33–4.00 |
| BER (backward extrusion ratio) | 1.20–4.00 |

## 3. Results

### 3.1. Forming Limit Diagram

A forming limit diagram was constructed from the result of the CFBE experiment, as shown in Figure 4. This diagram contains three zones, namely "safe forming", "surface-crack" and "piercing" zones. The surface-crack zone is divided into a forward surface-crack zone and a backward surface-crack zone; the piercing zone is divided into a forward piercing zone and a backward piercing zone. The forming limit diagram shows that safe forming appears when both the FER and BER are large. Surface-cracks occur when the FER

or BER is reduced. The cracks would occur on the outer surface of the forward extruded pin when the FER is larger than the BER. Conversely, the cracks would occur on the inner surface of the backward extruded cup when the FER is less than the BER. Piercing occurs when the FER or BER is further reduced. It would start from the outer corner of the forward extruded pin to the inner corner of the backward extruded cup when the FER is larger than the BER. Conversely, piercing would start from the inner corner of the backward extruded cup to the outer corner of the forward extruded pin when the FER is less than the BER. The insets show the outlook of the workpiece from the selected forming conditions.

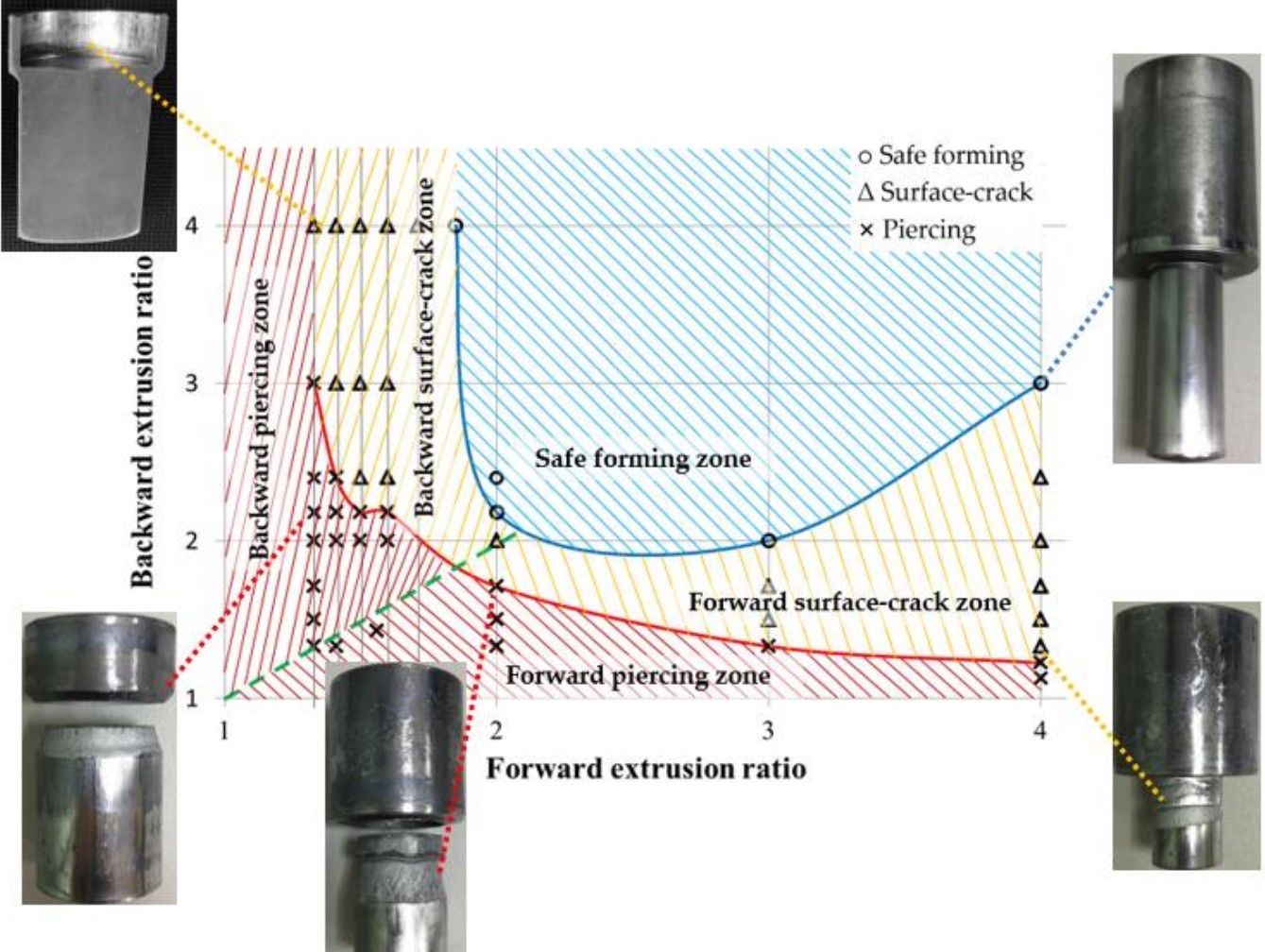

**Figure 4.** Forming limit diagram of CFBE experiment.

### *3.2. Forming Load-Stroke Curves*

#### 3.2.1. Forward Defects

Figure 5a shows the photos of the representative forming defects for the BER ranging from 1.2 to 2.4 and the FER fixed at 4. Piercing defects appear at low BERs for 1.20 and 1.26. The forming defects transit to surface-cracks on the outer surface of the forward extruded pin when the BER is increased to 1.33. The surface-cracks become spotty and imperceptible when the BER is increased to 2.4, which makes it subjective in drawing the border line between the crack zone and the safe forming zone in Figure 4.

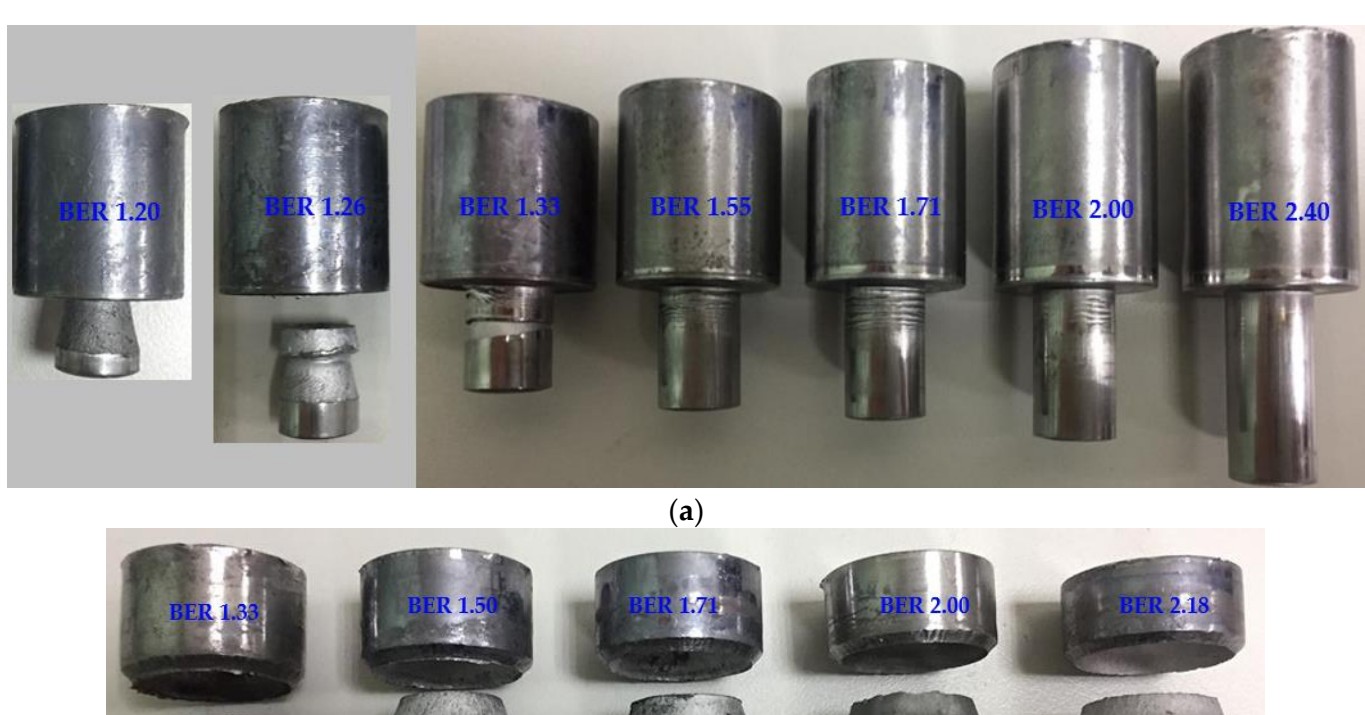

(a)

(b)

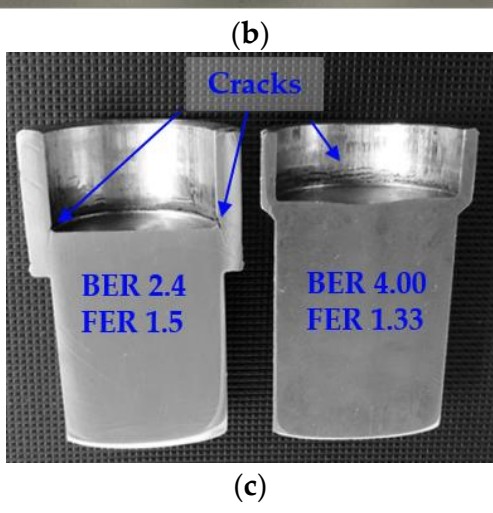

(c)

**Figure 5.** Photos of representative forming defects of various combinations of FER and BER: (**a**) Forward piercing and surface-crack defects for FER fixed at 4, (**b**) backward piercing defects for FER fixed at 1.33, (**c**) backward defects for FER 1.5-BER 2.4 and FER 1.33-BER 4.

The corresponding forming load-stroke curves for the CFBE experiment used in Figure 5a are shown in Figure 6a. This shows the load level increases with the BER. The forming load exhibits a deep drop when piercing occurs as the stroke reaches around 16 mm at low BERs for 1.20 and 1.26. The deep drop is followed by a short duration of forming load and this corresponds to the secondary forming of the short extruded pin connected to the fractured cone-part. There is a slight dip observed in the load-stroke curve for the case of BER equal to 1.33. This corresponds to a large surface-crack formed on the

outer surface of the extruded pin. When the BER is further increased, the cracks become less serious and no obvious dip is observed in the load-stroke curves.

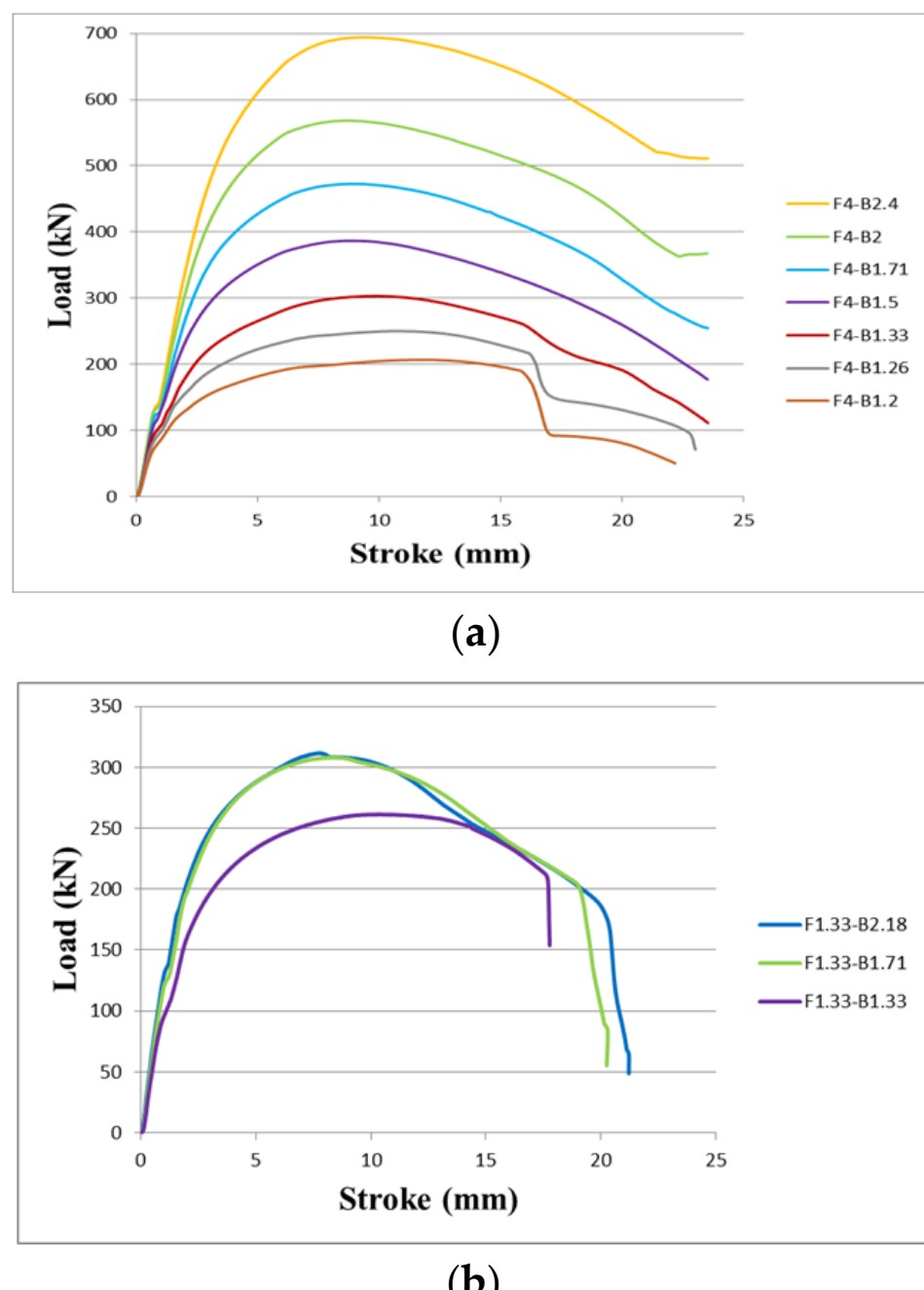

**Figure 6.** Forming load-stroke curves for (**a**) FER fixed at 4, and (**b**) FER fixed at 1.33.

### 3.2.2. Backward Defects

Figure 5b shows the photos for the BER ranging from 1.33 to 2.18 and the FER fixed at 1.33. Piercing defects appear for all these low FER conditions. Some corresponding forming load-stroke curves are shown in Figure 6b. They show that the load level increases with the BER and exhibits a deep drop when piercing occurs at the final stroke. The length of stroke increases as the BER increases, indicating the increase of the constraint from the extrusion chamber to suppress the occurrence of piercing. Therefore, unlike Figure 6a, there is no short duration of forming load after the drop and the secondary forming stage was not

observed. This is attributed to the overall extrusion ratio (i.e., the sum of FER and BER) is less than those used in Figure 6a.

Figure 5c shows the cross-section of the workpieces with backward surface-crack defects on the inner surface of the backward extruded cup. For a low combination extrusion ratio of FER 1.5 and BER 2.4, the surface-crack appears on the cup corner and is ready to propagate toward the pin. This is also evidence that the piercing defect initiates from the backward extruded cup-corner to the forward extruded pin-corner, when the FER is less than the BER. For a higher combination extrusion ratio of FER 1.33 and BER 4, the surface-cracks appear near the cup corner, and the tendency for piercing is suppressed.

### 3.2.3. Comparison of the Forming Load between the Experiment and the Simulation

Figure 7 shows the forming load-stroke curves obtained from the DEFORM simulation by using the normalized Cockcroft and Latham damage criterion. Since the material was assumed to be plastic, the elastic part at the starting stroke does not appear in the curves. The load levels are similar to those obtained from the experiment as shown in Figure 6. However, for FER fixed at 4, the simulation did not agree well near the final stroke when serious cracks occur for BER 1.33 or piercing occurs at BER 1.26, as shown in Figure 7a. Additionally, for FER fixed at 1.33, the simulation failed to predict when piercing occurs at BER 1.33, as shown in Figure 7b. This indicates that no serious fracturing occurs at this condition.

### 3.3. Simulation of the Fracturing Damage

The normalized Cockcroft and Latham damage criterion was used in determining the level of ductile fracture, and scoping the occurrence of forming fractures. Figure 8 shows the distribution of the damage values from the DEFORM 2D simulation. Figure 8a shows that large damage values appear on the outer surface of the forward extruded pin when the FER is larger than the BER. In practice with the CFBE, the occurrence of piercing or surface-crack defects might be misinterpreted as a "hole-punching effect" when the sizes of the punch and die outlet are similar.

The maximum damage value will shift to the middle of the diagonal connecting the pin-and-cup corners when the FER is equal to the BER, as shown in Figure 8b. The direction of the propagation of cracks or piercing depends on the conditions of punch radius, die-outlet radius and lubrication which favor the forward or backward defects.

The large damage values will appear on the inner surface of the backward extruded cup when the FER is less than the BER, as shown in Figure 8c. The overall extrusion ratio is smaller than that of Figure 8a. There are also some high damage values appearing on the outer surface of the forward extruded pin, which promotes the occurrence of the piercing defects. This also helps explain why there is no secondary forming stage observed in Figure 6b due to an insufficient constraint from the extrusion chamber. The sizes of the punch and die outlet are also similar, but the cracks start from the cup corner instead of the pin corner. This contradicts the common interpretation of the forming fracture with the "hole-punching effect" the cracks of which should initiate at the pin corner.

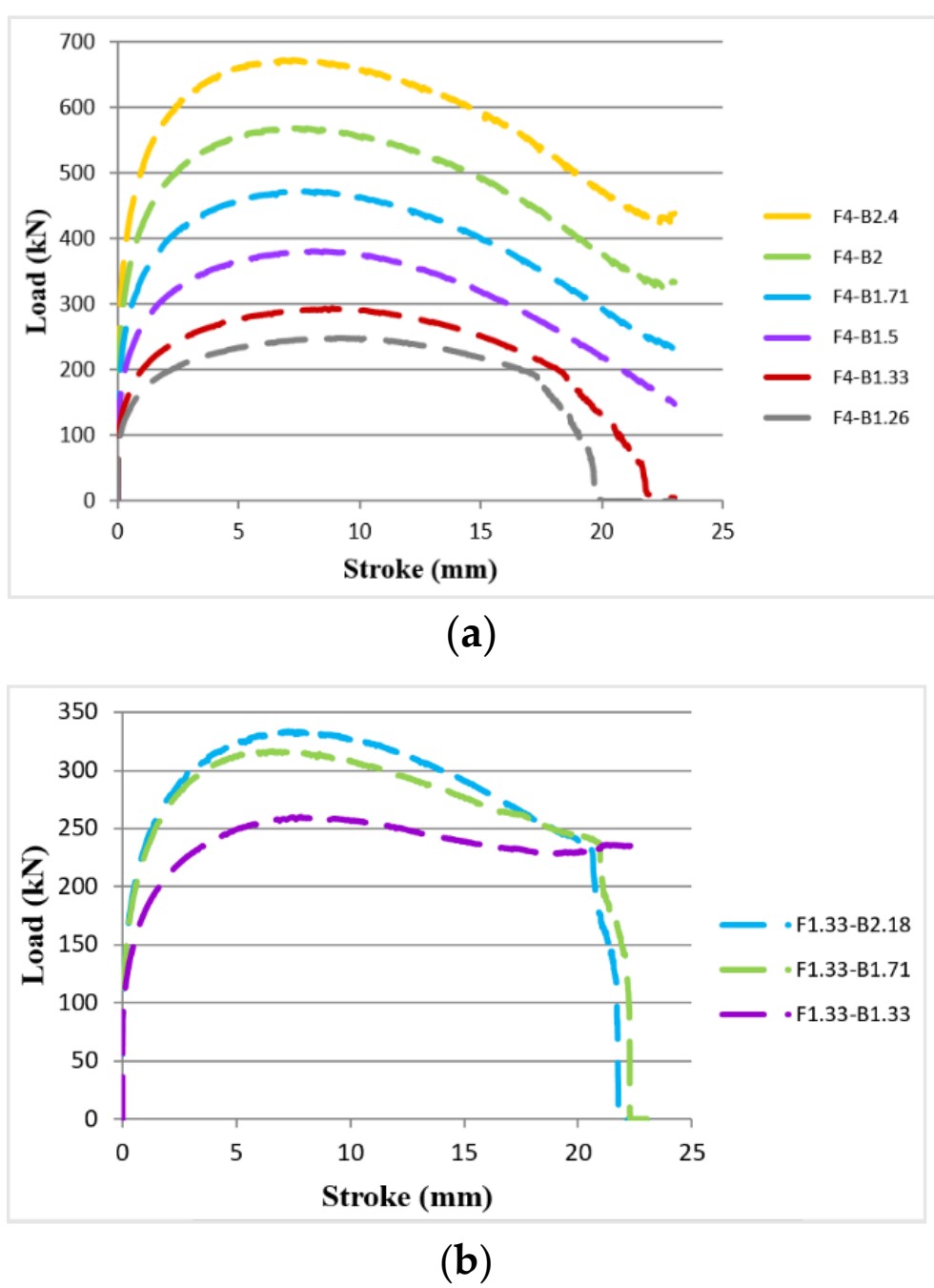

**Figure 7.** Forming load-stroke curves from DEFORM simulation for (**a**) FER fixed at 4, and (**b**) FER fixed at 1.33.

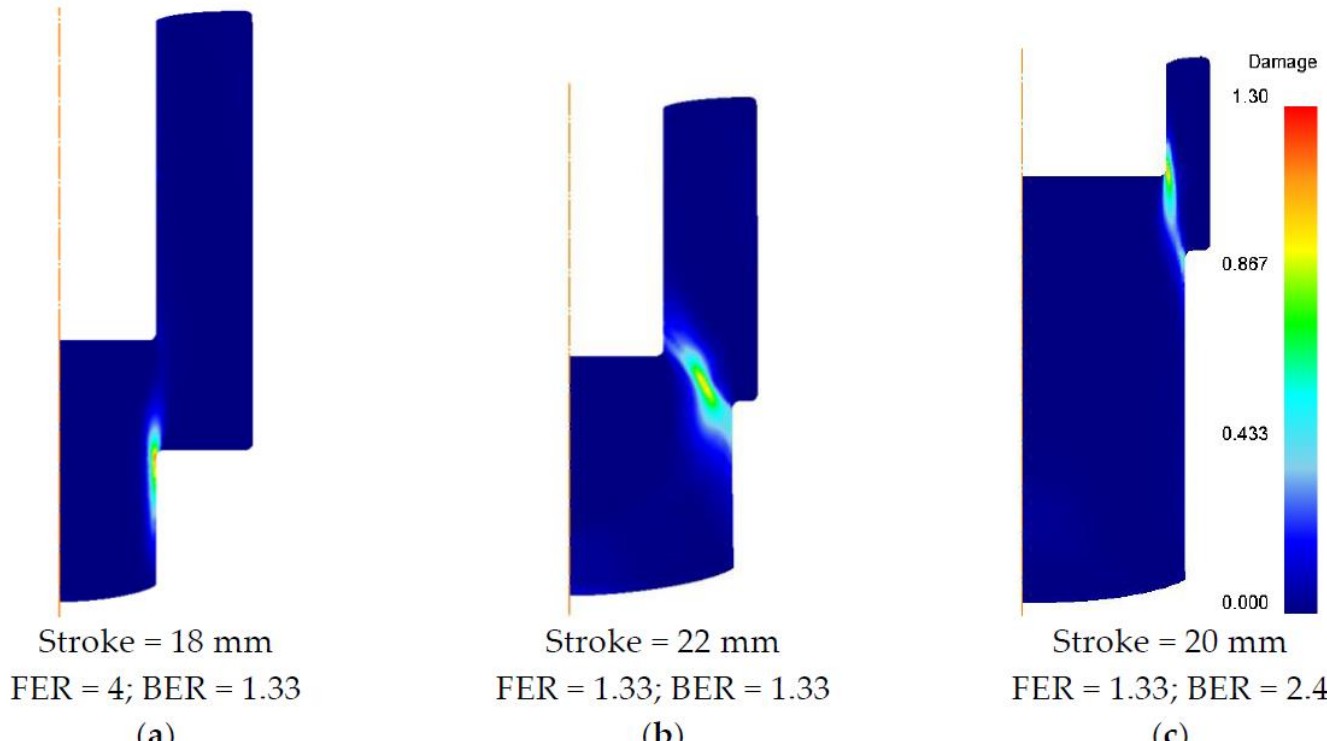

**Figure 8.** Distribution of the damage values from DEFORM 2D simulation of CFBE experiment: (**a**) FER = 4; BER = 1.33, at stroke 18 mm, (**b**) FER = 1.33; BER = 1.33, at stroke 22 mm, (**c**) FER = 1.33; BER = 2.4, at stroke 20 mm.

## 4. Discussion

A forming limit diagram, as shown in Figure 4, was constructed from the CFBE experiment. This provides a conception in choosing appropriate CFBE ratios in forming fasteners featuring a forward extruded pin and a backward extruded cup. More experiments are required because there are still some border-lines that have not been clearly defined. A separate unidirectional extrusion scheme is recommended to avoid the occurrence of the forming defects, when the geometry is determined to be unsafe from the forming limit diagram for the CFBE [1].

The current forming limit diagram is available for the cold forming of structural steel. The diagram needs further adjustment for other metals of different ductility, or different lubrication conditions. In the use of hot forming, additional parameters such as temperature and punch speed have to be considered [8].

The finite element analysis provided in Figure 8 was able to scope the positions with high damage values but with fair accuracy. The occurrence of cracks and piercing from the experiment does not exactly correspond to the critical damage value from the computation of finite element simulation. Some other damage criteria should be tested [10,11], in order to find a more suitable criterion for the CFBE process.

## 5. Conclusions

A series of the CFBE tests with various combinations of the FER and the BER were conducted. A forming limit diagram, detailed with the piercing and surface-crack defects on the forward extruded pin or the backward extruded cup, was constructed. With the aid of the forming load-stroke curves and the finite element analysis of fracture damage, the following conclusions can be drawn.

Piercing defects appear in the CFBE of extremely low FER and BER. The forward piercing occurs when the FER is greater than the BER. The piercing fracture propagates from the forward extruded pin-corner to the backward extruded cup-corner. Conversely, the backward piercing occurs when the FER is less than the BER. The piercing fracture

propagates from the backward extruded cup-corner to the forward extruded pin-corner. The load level is low and the forming load-stroke curves exhibit a sharp drop when piercing occurs.

Surface-crack defects occur when the FER and BER are slightly increased. The forward surface defects occur when the FER is greater than the BER, and the cracks appear on the forward extruded pin-surface. Conversely, the backward surface defects occur when the FER is less than the BER, and the cracks appear on the backward extruded cup-surface. The load level increases and the forming load-stroke curves might exhibit a slight dip when serious surface-cracks occur.

The safe forming zone would result when both the FER and BER are further increased substantially. The tendency of crack propagation would be suppressed by the high pressure of the extrusion chamber, and the forming loads are typically high.

**Author Contributions:** Conceptualization, H.-S.L.; methodology, H.-S.L.; software, W.-S.L.; validation, W.-S.L.; writing—original draft preparation, C.-Y.L. All authors have read and agreed to the published version of the manuscript.

**Funding:** This research was funded by the Ministry of Science and Technology of Taiwan, ROC, grant number MOST 106-2622-E-992-307-CC3.

**Institutional Review Board Statement:** Not applicable.

**Informed Consent Statement:** Not applicable.

**Data Availability Statement:** Not applicable.

**Acknowledgments:** The joint financial support by Wei Zai Industry is acknowledged.

**Conflicts of Interest:** The authors declare no conflict of interest.

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
