# Peer review of "Piercing and Surface-Crack Defects in Cold Combined Forward-Backward Extrusion"

_applsci, doi:10.3390/app11093900_

Round 1

Reviewer 1 Report

The author can be more specific about the specific applications of the fasteners in automotive and other industries. The author cited the references, but can elaborate more on the current processes (example: thread rolling) and other extrusion processes broadly used in forming fasteners [which ones? for example: engine bolts? connecting rod bolts?, etc., as cited in ref 1], and automotive parts [what are these? you only cited ref 2]. So, be specific and show sketches of those. Write a comparison of issues and failures of manufacturing fasteners by other means with the proposed method. Is this hot forming or cold forming. How does the quality compare between the two?

Deform 3D software is used for the simulation. Are there papers that used Autoform or LS-Dyna, Ansys, Abaqus, etc.? Can you briefly discuss and write a comparison between these simulations?

How about the power law used by other researchers? Write briefly about that too and include all others works in the list of references.

Reviewer 2 Report

Overall, the research is well conducted and the results are very interesting. I got several suggestions which I believe will help to improve the paper:

Abstract, line 13, combination should be combined.

The research background and literature review in the introduction are not sufficient, in which only 4 references exist. Extrusion process has been widely used in forming lightweight sections/profiles for potential application in automotive parts. It would be helpful to add: 10.1016/j.ijmecsci.2018.02.028; 10.1016/j.proeng.2017.10.999; 10.1016/j.ijlmm.2018.03.004

More details are needed for the FE modelling using Deform. e.g. The geometry model and finite element model.

The reason why 2D was used should be justified.

The FE model should be validated by comparing the simulated forces with experimental ones.

For the cold forming, the elastic deformation of the billet and die should have an effect on the forming process, i.e. springback. The reason why the workpiece was assumed to be plastic and the punch and die were assumed to be rigid should be justified.

Why was the normalized Cockcroft and Latham damage criterion used? How was it used in Deform?

Round 2

Reviewer 2 Report

Thank you for the response, I have no further comments.